# Novel Pyrrolidinium-Functionalized Styrene-b-ethylene-b-butylene-b-styrene Copolymer Based Anion Exchange Membrane with Flexible Spacers for Water Electrolysis

**DOI:** 10.3390/membranes13030328

**Published:** 2023-03-13

**Authors:** Ziqi Xu, Sofia Delgado, Vladimir Atanasov, Tobias Morawietz, Aldo Saul Gago, Kaspar Andreas Friedrich

**Affiliations:** 1German Aerospace Center (DLR), Institute of Engineering Thermodynamics, Pfaffenwaldring 38-40, 70569 Stuttgart, Germany; 2Laboratory for Process Engineering, Environmental, Biotechnology and Energy (LEPABE), Faculty of Engineering of University of Porto, Rua Roberto Frias S/n, 4200-465 Porto, Portugal; 3Institute of Chemical Process Engineering, University of Stuttgart, Boeblinger Strasse 78, 70199 Stuttgart, Germany; 4Faculty of Science, Energy and Building Services, Esslingen University of Applied Sciences, Kanalstraße 33, 73728 Esslingen am Neckar, Germany; 5Institute of Building Energetics, Thermal Engineering and Energy Storage (IGTE), University of Stuttgart, Pfaffenwaldring 6, 70569 Stuttgart, Germany

**Keywords:** anion exchange membrane, anion exchange membrane water electrolysis, flexible oxide spacers, conductivity, chemical stability

## Abstract

Anion exchange membranes (AEM) are core components for alkaline electrochemical energy technologies, such as water electrolysis and fuel cells. They are regarded as promising alternatives for proton exchange membranes (PEM) due to the possibility of using platinum group metal (PGM)-free electrocatalysts. However, their chemical stability and conductivity are still of great concern, which is appearing to be a major challenge for developing AEM-based energy systems. Herein, we highlight an AEM with styrene-b-ethylene-b-butylene-b-styrene copolymer (SEBS) as a backbone and pyrrolidinium or piperidinium functional groups tethered on flexible ethylene oxide spacer side-chains (SEBS-Py2O6). This membrane reached 27.8 mS cm^−1^ hydroxide ion conductivity at room temperature, which is higher compared to previously obtained piperidinium-functionalized SEBS reaching up to 10.09 mS cm^−1^. The SEBS-Py206 combined with PGM-free electrodes in an AWE water electrolysis (AEMWE) cell achieves 520 mA cm^−2^ at 2 V in 0.1 M KOH and 171 mA cm^−2^ in ultra-pure water (UPW). This high performance indicates that SEBS-Py2O6 membranes are suitable for application in water electrolysis.

## 1. Introduction

Transitioning to a net-zero world is one of the greatest challenges that humankind has faced in modern society. To fulfill this goal, the need of creating a cost-effective electrolysis method for green hydrogen production in power supply systems is highly emphasized by the demand in industry [1,2,3,4]. Compared with proton exchange membrane (PEM) electrolysis, alkaline water electrolysis (AWE) has risen to prominence because of its ability to use low-cost PGM-free catalysts and high industrial convertibility [5,6]. However, the high corrosive alkaline electrolyte (30 wt% KOH) is responsible for several issues, including carbonate formation, the need for electrolyte circulation, and gas-emitting impurities [7,8].

Anion exchange membranes (AEM) are regarded as potential solutions to these challenges since they facilitate the reduction of electrolyte concentration [6,9,10,11,12,13,14] and improve gas purity and electrical efficiency. However, no dependable commercial AEMs with the requisite conductivity and stability exist [15,16,17,18], posing considerable obstacles in the development of high-performance AEMWEs. Due to the hydroxide’s strong nucleophilic and basic character, the chemical stability in an alkaline environment at elevated temperature is the most essential challenge for AEM. In detail, for the backbone of AEM, heteroatoms, e.g., oxygen and sulfur, must be avoided because they can be quickly destroyed by the hydroxide [19]. When it comes to the stability of functional groups on quaternary ammonium (QA), there are many degradation processes that include nucleophilic substitution, elimination (Hofmann elimination), and rearrangement reactions [20,21]. Marino and Kreuer [22] evaluated the stability of a range of functional groups against nucleophilic attack, including 1,4-diazabicyclooctane (DABCO), imidazolium, and methyl piperazine. Due to their unfavorable bond angles and lengths in the chemical transition states, the 6-membered and 5-membered cycloaliphatic QAs have exhibited good resistance to nucleophilic substitution and elimination.

The moderate conductivity of the AEMs is also a hurdle for their performance [17,23]. To improve ionic conductivity, several effective approaches have been investigated. Although an increase in ion exchange capacity (IEC) can contribute significantly to an increase in conductivity, an overabundance of IEC can result in uncontrolled swelling and membrane instability. Constructing a phase separation by introducing comb-shaped structures [24,25] while simultaneously increasing the number of cations per chain [26,27] is a more efficient technique for conductivity improvement without compromising dimensional stability.

A piperidinium/pyrrolidinium-functionalized multi-cation comb-shaped polymer structure is discussed in this paper. Because of its high stability and mechanical integrity, a multi-block copolymer SEBS was selected as the backbone. More hydrophilic ethylene glycol-based side chains are also added to achieve a more flexible spacer structure and induce microphase separation [28]. As a result, a membrane with piperidinium/ pyrrolidinium functionality and flexible ethylene oxide spacers has good conductivity and stability [29,30,31,32]. Different functional groups (piperidinium/pyrrolidinium) and architectures of the side chain, such as the pre-functionalized end of the spacer, are discussed in detail. The properties of the membrane, such as conductivity, IEC, and alkaline stability were studied.

## 2. Materials and Methods

### 2.1. Materials

SEBS (30 wt% styrene, Taipol 6152) was provided by TSRC cooperation. Commercial catalysts H2GEN-M (which contains Mo and C) and OXYGN-N (which mainly contains Ni and Fe) from CENmat were used for fabrication of the cathode and anode, respectively, for the AEM electrolyzer tests. These catalysts are platinum group metal (PGM)–free. Chlorotrimethylsilane, 1,3-dibromopropane, 1,4-dibromobutane, sodium chloride, and potassium hydroxide were purchased from TCI Europe. Trioxane, tin (IV) chloride, 1,6-dibromohexane, chloroform, xylene, dimethyl sulfoxide, water solution of trimethylamine (33%), ethanol, dichloromethane, and tetrahydrofuran were all purchased from Merck. All chemicals were used as received without further purification.

### 2.2. Synthesis and Membrane Preparation

#### 2.2.1. Synthesis of Chloromethylated SEBS (cmSEBS)

The cmSEBS synthesis in detail has already been published elsewhere [33]. In an argon-purged three-neck round-bottom flask, chloroform (250 mL) was added to SEBS polymer (4 g, 25 mmol). Then, the mixture was stirred at room temperature for 4 h. After the polymer dissolution, trioxane (5.4 g, 60 mmol) was added and the flask was placed in an ice bath until the temperature of the mixture dropped down to 1 °C. Chlorotrimethylsilane (22.8 mL, 37.5 mmol) was injected to the reaction mixture and subsequently tin (IV) chloride (3 mL, 3 mmol) into the reaction mixture. The mixture was stirred at a temperature at less than 5 °C for 30 min and then at room temperature for 24 h. Afterwards, the mixture was transferred into a beaker filled with 300 mL of ethanol/water (50 % *v/v*) mixture to terminate the reaction at the end of the reaction time. The product was twice precipitated from chloroform solution into ethanol/water (50 % *v/v*) mixture after filtering to collect the solid. The final product was dried in a vacuum oven at 50 °C overnight. The ^1^H NMR spectrum was shown in Figure 1a.

#### 2.2.2. Synthesis of 1,6-Bis(piperidin-1-yl) Hexane (P2C6) and 1,6-Bis(pyrrolidin-1-yl) Hexane (Py2C6)

Piperidine (26 mL, 250 mmol) was dissolved in 150 mL acetone under argon and stirred for 30 min at room temperature. 1,6-dibromohexane (16 mL, 100 mmol) was added dropwise to the piperidine solution after cooling in an ice bath at T < 5 °C. The reaction mixture was then stirred at room temperature for 48 h after adding anhydrous potassium carbonate (20 g, 152 mmol). Once removing the undesirable precipitates using filtration (solid phase removed), the solvent was removed using vacuum evaporation. To extract the product, 60 mL dichloromethane was added to the residue, which was then rinsed with deionized water (5 × 100 mL) and dried at 60 °C for 10 h, leaving a thick yellowish liquid (16.4 g, about 65 mmol). Figure 2d shows the ^1^H NMR spectrum of P2C6.

For the synthesis of Py2C6, the same synthetic procedure as for P2C6 was followed using pyrrolidine instead of piperidine. The ^1^H NMR spectrum is shown in Figure 2b.

#### 2.2.3. Synthesis of 1,2-bis(2-piperidinylethoxy) Ethane (P2O6) and 1,2-Bis(2-pyrrolidinylethoxy) Ethane (Py2O6)

Piperidine (26 mL, 250 mmol) was dissolved in 150 mL acetonitrile under argon and stirred for 30 min at room temperature. 1,2-bis(2-chloroethoxy)ethane (14 mL, 100 mmol) was then mixed dropwise to piperidine solution. After that, anhydrous potassium carbonate (20 g, 0.145 mol) was added and stirred for 48 h at 60 °C. After removing the undesirable precipitate using filtration (solid phase removed), 100 mL water was added in. Later, the solution was treated with ethyl acetate to extract the product and ethyl acetate phase was obtained. Rotating evaporation was taken afterwards to remove the ethyl acetate. Finally, the remainder was put into the vacuum oven at 60 °C for 24 h, yielding a viscous dark yellow liquid (9.4 g, about 33 mmol). Figure 2e shows the ^1^H NMR spectrum of P2O6.

For the synthesis of Py2O6, the same synthetic procedure as for P2O6 was followed using pyrrolidine instead of piperidine. The ^1^H NMR spectrum are shown in Figure 2c.

#### 2.2.4. Membrane Preparation

Different membranes structures obtained in this study were shown in Figure 2 and the overall membrane preparation process in Figure 1. For SEBS-P2C6 membrane preparation, cmSEBS (0.7 g) was dissolved into 25 mL chloroform at 50 °C for 2 h to form nearly 2 wt% solution. Then 1 mL of P2C6 (around 0.003 mol) was added in 25 mL chloroform and 0.2 mL iodomethane dropped in, with continued stirring for 2 h to have mP2C6. Subsequently, two solutions mixed together, and the membrane was cast from the solution on a PTFE square dish (15 cm × 15 cm) and dried first at 50 °C for 4 h and then 100 °C for 20 h in vacuum. Finally, the SEBS-P2C6 membrane was detached from the PTFE dish. SEBS-P2C6 membrane was boiled in water to remove the remaining solvent. For SEBS-Py2C6, SEBS-P2O6, and SEBS-Py2O6, preparation methods were similar to that of SEBS-P2C6 but used Py2C6, P2O6, or Py2O6 instead of P2C6.

### 2.3. Instrumental

#### 2.3.1. NMR Spectroscopy and Fourier-Transform Infrared Spectroscopy (FT-IR)

Bruker Avance III HD 400 NanoBay NMR spectrometer was used to measure the ^1^H and ^13^C NMR spectroscopy of the polymer at room temperature in either DMSO-d_6_ or CDCl_3_ as solvents, with tetramethylsilane as the internal reference (TMS). A Nicolet iS5 (Thermofisher Scientific, Karlsruhe, Germany) and a diamond attenuated total reflectance (ATR) module was used to record FT-IR spectra of the membranes with 64 scans in the wave number range of 4000 to 400 cm^−1^.

#### 2.3.2. Atomic Force Microscope (AFM)

Icon XR (Bruker Karlsruhe, Germany) was used to perform the atomic force microscopy on in PeakForce Tapping Mode to measure the nanomechanical properties of the membranes. At each measurement point, the AFM modus records and evaluates force-distance curves. Along with height adhesion, stiffness and deformation were measured at the same time. Bruker Scanasyst-Air tips (k = 0.4 N m^−1^) with a tip radius of 2 nm were used. The membranes were glued to 15 mm AFM steel-discs with adhesive carbon tape (Plano) and measured at ambient conditions. For all membranes, the image size for the AFM test was set to 1 µm and they were measured at 0.977 Hz with 512 × 512 pixels.

#### 2.3.3. Membrane Conductivity

A Teflon contact cell and four electrode methods were used to measure the through plane conductivity of the membrane. Electrodes are made of Au and a Zahner-elektrik IM6 device (Zahner-elektrik GmbH, Kronach, Germany) was used to record the electrochemical impedance spectroscopy by measuring membrane resistances in 1 M NaCl solution or 1M KOH at room temperature. In a frequency range of 200 KHz–8 MHz, by intersecting the impedance with the true x-axis, the resistance was calculated. Since we do not have a setup to remove CO_2_, OH^−^ conductivity was measured immediately after the membrane was changed to OH^−^ form. For each membrane, four samples were collected. The conductivity was calculated by the equation below (Equation (1)).
(1)$$\sigma =\frac{1}{R_{sp}}=\frac{d}{R\times A}$$
where *σ* is the conductivity (S cm^−1^), *R_sp_* is the resistivity (Ω cm), *d* is the thickness of membrane (cm), *R* is the ohmic resistance (Ω), and *A* is the electrode area (cm^2^)

#### 2.3.4. Water Uptake and Swelling Ratio

The weight and dimension differences of membranes after soaking in deionized water for 48 h at room temperature and drying in a vacuum oven at 60 °C for 24 h are measured to calculate the water uptake (*WU*) and swelling ratio (SR) in Cl^−^ form. The WU is calculated by the Equation (2) as follows:(2)WU%=(mwet−mdry)mdry×100
where *m_wet_* and *m_dry_* are the weight of wet and dry membranes in Cl^−^ forms in grams, respectively.

The *SD* was calculated by the Equation (3) as follows:(3)SD%=(lwet−ldry)ldry×100
where *l_wet_* and *l_dry_* are the geometric length of the wet and dry membranes in Cl^−^ forms, respectively.

#### 2.3.5. Ion Exchange Capacity (IEC)

IEC was determined by a back-titration strategy. The membrane sample was immersed in 1M NaOH solution, then washed with DI water before being immersed in a soaked sodium chloride arrangement at room temperature for 1 day. Afterwards, membranes were taken out from the solution. Hydrochloric acid (3 mL, 0.1 M) standard arrangement was added and the mixture was stirred at room temperature for 1 day. The following arrangement was titrated with a standard 0.1 M sodium hydroxide solution. The membrane was thoroughly washed with DI water and dried overnight at 90 °C. On a balance, the dry weight of the membrane was determined. The *IEC* was calculated by Equation (4).
(4)$$IEC=\frac{C_{HCl}\times V_{HCl}-C_{NaOH}\times V_{NaOH}}{m_{dry}}$$
where *IEC* is the ion exchange capacity (Cl^−^ form, mmol/g), *C_HCl_* is the concentration of the hydrochloric acid solution (mmol/mL), *V_HCl_* is the volume of the hydrochloric acid solution used (mL), *C_NaOH_* is the concentration of the sodium hydroxide solution (mmol/mL), *V_NaOH_* is the added volume of the sodium hydroxide solution (mL), and *m_dry_* is the dry weight of the membrane (g).

#### 2.3.6. Chemical Stability of the Membranes

Chemical stability was determined in 1 M KOH at 90 °C for 30 days. Several pieces of membrane were washed repeatedly with water for 1 day before being put into the alkaline solution at 90 °C in an oven. Every 5 days during the test, a small piece of membrane was cut to test the conductivity, and new KOH solution was exchanged. Following that, the membranes were thoroughly washed with deionized water before being tested for OH^−^ conductivity. The conductivity retention rate (CR%) of the membrane was calculated using the following equation.
(5)$$\mathrm{CR}\%=\frac{\sigma_{1}}{\sigma }\times 100$$
where σ is the conductivity of the membrane before treatment in KOH.

### 2.4. Fabrication of Membrane Electrode Assemblies (MEAs) for AEMWEs

Catalyst coated substrates (CCS) were prepared by dispersing the H2GEN-M and OXYGN-N catalysts on carbon paper substrates from Sigracarb using a manual spraying technique [34]. Initially, chloromethylated SEBS (cmSEBS) was used as the ionomer precursor and dissolved in tetrahydrofuran.

Thereupon, the catalyst suspensions were prepared using isopropanol (Merck, Darmstadt, Germany) and ultra-pure water (Millipore, Burlington, MA, USA) as solvents and by maintaining an ionomer to catalyst ratio of 3:7, and 4 cm^2^ active area CCSs of 4 mg cm^−2^ (anode composed of OXYGEN-N and cathode composed of H2GEN-M catalyst) was fabricated and immersed into a 500 mL trimethylamine solution to functionalize the cmSEBS binder overnight. Then, CCSs were thoroughly washed with ultra-pure water and then submerged in a 1 M KOH solution for 15 h at room temperature to convert the chloride anion into the hydroxide anion. The membranes were finally rinsed with water prior to their assembly. A similar approach was performed to prepare CCS to be tested in UPW; for that, previous in situ activated CCS in 0.1 M KOH was flushed for half an hour in UPW until reaching a neutral pH at the outlet of the AEMWE. 

### 2.5. AEMWE Cell Test

The whole AEM electrolysis setup in this work was shown in Figure 2. The AEMWE cell consisted of titanium bipolar plates (BPPs) and porous transport layers (PTL) of Ti porous sintered layer (PSL) on Ti mesh (PSL/mesh-PTL) [35] (GKN Sinter Metals) and also carbon paper inside for both cathode and anode. The different cell assemblies were accomplished by cramming the CCSs together with the SEBS-Py206 membrane which was as well immersed for 15 h in 1 M KOH solution and then thoroughly rinsed with deionized water. A torque of 0.6 Nm was applied on 4 screws, which allowed closing of the cell [35]. The cell was then ready to test, and 1 L/min N_2_ was continuously bubbled into the electrolyte, either 0.1 M KOH or ultra-pure water, in a closed-loop system to remove the CO_2_ dissolved in the water and thus avoid the generation of precipitates and to maintain the chemical/mechanical integrity of the membrane. Nevertheless, contact with ambient air/CO_2_ cannot be entirely disregarded throughout the cell assembly preparation and activation steps. The electrochemical characterization was accomplished using a Zahner Zennium Pro electrochemical workstation (potentiostat/galvanostat) and a Zahner PP24 booster was employed to reach currents over 4 A. The cell was initially conditioned in situ in a 0.1 M KOH solution at 60 °C by monitoring the potential as function of the current density using a dwell time of 120 s and with small current increments of 50 mA cm^−2^ and then 150 mA cm^−2^ until reaching 1 A cm^−2^. Five polarization curves were retrieved in potentiostatic mode, from 1.3 V to 2.5 V using a scan rate of 20 mV s^−1^ in the 0.1 M KOH electrolyte at 60 °C. An electrochemical impedance spectrum (EIS) was recorded in galvanostatic mode at 200 mA cm^−2^ using an amplitude of 20 mA in the frequency range of 100 mHz to 1 kHz. The cell assembly was then purged with pure water flowing for 30 min and the electrochemical characterization steps were repeated at 60 °C using ultra-pure water (UPW) as electrolyte at neutral ph.

## 3. Results and Discussion

### 3.1. Synthesis and Preparation

In this paper, SEBS copolymer is chosen as the backbone for the preparation of the AEM. SEBS copolymer contains ethylene and butadiene as the flexible blocks as well as styrene as the rigid blocks. The polymer modification comprises two parts: chloromethylation and quaternization. For the chloromethylation, as shown in Figure 2a, the chloromethylation rate of SEBS is 83.5%. Subsequently, cmSEBS reacted with some side chain agents such as mP2C6 and mP2O6 (Figure 1) and the impact of side chain agents on the chemical stabilization via functionalization of the membrane, swelling changes and proton conductivity were studied in this work.

In detail, ^1^H NMR was recorded on cmSEBS to investigate the grafting rate and chemical composition. The resonance peak with chemical shift of 4.5 ppm belongs to -CH_2_Cl of benzyl chloride and the signals at 6.3–7.25 ppm belongs to aromatic protons of the styrene [36,37]. By comparing the ratio of these two peaks, -CH_2_Cl (peak 8) to Ar-H (peaks 1 and 2) in Figure 2a, 83.3% of the styrene part was chloromethylated (25.4% of the whole polymer).

For the structures of 1,6-bis(pyrrolidin-1-yl) hexane (Py2C6) and 1,6-bis(piperidin-1-yl) hexane (P2C6), the ^1^H NMR spectra chemical shifts are shown in Figure 2b,d and those for 1,2-bis(2-pyrrolidinylethoxy) ethane (Py2O6) and 1,2-bis(2-piperidinylethoxy) ethane (P2O6) are shown in Figure 2c,e. Due to the absence of CH_2_Br signals in all structures and the corresponding integral ratios between the N-CH_2_- and the rest of the protons in the structure, we are able to conclude the successful synthesis of those side-chain molecules (P2C6, P206, Py2O6 and Py2C6). 

After the functionalization, the corresponding membrane samples were not soluble in any common organic solvents. Therefore, the membranes were subjected to ATR-FTIR spectroscopy to find out whether the cmSEBS had reacted with the amine side chain (mPy2O6 and mP2O6). Taking SEBS-Py2O6 for example, in the ATR-FTIR spectrum of cmSEBS, a new peak at 725 cm^−1^ appeared in comparison to the pristine SEBS (Figure 3). This peak corresponds to C-Cl stretching vibration of the chloromethyl group. After the membrane was cast from the reaction mixture of cmSEBS, Py2O6 and iodomethane, a new peak appeared in the ATR-FTIR at 1120 cm^−1^. This peak is typical for C-O stretching vibration, which indicates the membrane functionalization.

### 3.2. Conductivities of Piperidinium/Pyrrolidinium Functionalized Membranes Based on SEBS

Commonly, SEBS-based piperidinium-functionalized AEMs have not shown high conductivity due to the relatively low degree of functionalization and respectively low IEC [38,39]. In the past, to improve the conductivity, several strategies have been tried, such as building a comb-shaped structure [40], having multi-cation side chains [41], and crosslinking with backbones being additionally functionalized [42]. Among them, using a comb-shaped structured polymers could induce formation of the micro-phase separation which often enhances the conductivity of the membrane. In our previous study, a piperidinium-functionalized flexible ethylene oxide spacer structure was applied as a crosslinker in the presence of trimethylamine to convert the chloromethyl group to a functional membrane [43]. However, this strategy also brought some drawbacks. Firstly, the crosslinkers reduce the IECs of the membranes rapidly and thus lose the advantage of the multi-cation side-chain structure. Secondly, functionalization by post treatment may decrease the mechanical stability of the membrane and make the membrane brittle. And third, benzyl ammonium group is known to have inferior stability which may lead to the degradation in alkaline solution. Therefore, in this work we chose a functionalized flexible ethylene oxide spacer structure with a pre-functionalized end (of which one end of the diamines was quaternized by iodomethane before) and removed all the benzyl ammonium groups in the structure. As is shown in Figure 1, the first diamines (P2C6 and P2O6) were reacted with iodomethane to get unsymmetrical one-end quaternate amine-ammonium precursors (pre-functionalized end) (mP2C6/mP2O6). Subsequently, both cmSEBS and mP2C6/mP2O6 were dissolved to prepare the final membrane. Some of the most important properties relating to their application are shown in Table 1. The IECs of the membranes SEBS-P2C6 and SEBS-P2O6 are higher than the membranes SEBS-P2C6-TMA and SEBS-P2O6-TMA. However, the IECs of SEBS-P2C6 and SEBS-P2O6 are much lower than the fully functionalized membranes (IEC in theory is 2.7 mmol g^−1^), which is probably due to charge repulsion between the charged side chains and the charged polymer product, even though the amine-ammonium precursors are in excess. Due to the higher IEC, the conductivity of the SEBS-P2C6 and P2O6 improved a lot, reaching 17.5 mS cm^−1^ and 25 mS cm^−1^ at RT, which is higher than SEBS-based piperidinium-functionalized membrane published before (piperidinium functionalized SEBS-pi-73%, 10.09 mS cm^−1^) [1]. In all cases, Cl^−^ conductivity is about twice as low as than the OH^−^ conductivity, which is due to the larger and heavier Cl^−^ ion compared to OH^−^. This factor is slightly lower than the Cl^−^ and OH^−^ conductivities (σ_OH_^−^/σ_CL_^−^~2.7) that have been reported in the literature [44]. However, the residual CO_2_ present in our case may account to this difference for the OH^−^ value, although the test was performed with care immediately after removal from KOH solution.

Basically, pyrrolidinium is believed to be a very chemically stable functional group in alkaline solution because of its low ring strain and relatively high energy barrier towards E2 elimination [22]. Therefore, in this work, SEBS membranes having pyrrolidinium functional groups were synthesized and studied. In Table 1, SEBS-Py2C6 and SEBS-Py2O6 have shown relatively high conductivity being 17.6 mS cm^−1^ and 27.8 mS cm^−1^ at RT, respectively. Compared with piperidinium-functionalized membranes (SEBS-P2C6 and SEBS-P2O6), at almost the same IEC, pyrrolidinium-functionalized membranes showed similar conductivities, making them a promising candidate for AEM electrolysis application.

### 3.3. Membranes Morphology Study on SEBS-P2O6 and SEBS-Py2O6

In our previous work, the micro-phase separation of the ethylene oxide spacer structure in crosslinkers was investigated. In the AFM image, the membrane with the ethylene oxide spacer structure showed broader and better connected hydrophilic domains, which was believed would promote better formation of counter ion channels, thus improving the conductivity of membranes which have the ethylene oxide spacer structure crosslinkers [43]. In the literature, other studies were also taken to explain the high conductivity of the ethylene oxide spacer structure [28,32]. Similarly, in this work, membrane morphology was proven by AFM imaging in DMT modulus (Figure 4). The bright (stiffer) regions correspond to hydrophobic polymer backbone agglomeration, while the dark regions stand for hydrophilic side chains. For both SEBS-Py2O6 and SEBS-Py2C6 membranes, a clear phase-separation of hydrophilic and hydrophobic phases was observed. For SEBS-Py2O6 (Figure 4a,b), containing long flexible and hydrophilic side chains, the hydrophilic domains formed were much better connected and broader compared with SEBS-Py2C6 (Figure 4c,d). The broader and better interconnected hydrophilic domains of SEBS-Py206 make the ion-channels much better intercalated with relatively constant size, which enables efficient ion transport [45,46].

### 3.4. Chemical Stability of SEBS-Based Piperidinium/Pyrrolidinium-Functionalized Membranes

In order to check the alkaline stability, the membranes were immersed into1M KOH at 90 °C for more than 700 h. As is shown in Figure 5, AEMs with benzyl trimethyl ammonium groups showed low conductivity retention, which can be ascribable to the degradation of benzyl trimethyl ammonium in hot alkaline environment. At the same time, when adding ethylene oxide spacers to the side chain, there are no significant losses on membrane stability (SEBS-P2C6 and SEBS-P2O6), which has similar results in the literature [32]. Comparing the membranes having piperidinium groups with those having pyrrolidinium groups, after immersing AEMs in 1M KOH at 90 °C for more than 700 h, the conductivity of both SEBS-P2O6 and SEBS-Py2O6 still have nearly 95% conductivity retention, and we can draw the conclusion that replacing the piperidinium with pyrrolidinium cannot significantly change the chemical stability of the membrane. For SEBS-Py2O6, the conductivity only dropped from 27.8 mS cm^−1^ to 26.4 mS cm^−1^ (OH^−^ form), which may due to some benzyl group degradation in harsh hot alkaline environment or some radical attack [20,47]. Considering both the conductivity and stability of the membranes functionalized by piperidinium and pyrrolidinium functional groups suited on flexible ethylene oxide spacer side chains, a well-balanced conductivity and stability have been achieved.

### 3.5. AEM Water Electrolysis Measurements

After balancing the conductivity and chemical stability of the membranes, membranes with ethylene oxide spacer side chains and piperidinium/pyrrolidinium functional groups are necessary to achieve high performance in AEMWE. In this work, SEBS-Py2O6-based membranes and the ionomer were tested in a AEMWE cell using the OXYGN-N and H2GEN-M catalysts for the anode and cathode, respectively. As shown in Figure 6a, the cell performance at 60 °C in 0.1 M KOH was measured via polarization curves after performing a potentiostatic conditioning; see Appendix A. A current density of 520 mA cm^−2^ was reached at 2 V, which unveils a potential for future applications of the SEBS-Py2O6 membrane and SEBS-TMA ionomer for AEMWE with PGM-free catalysts. In fact, polyethylene-based membranes were attempted in similar AEMWE configurations (0.1 M KOH and 60 °C), achieving ca. 0.1 A cm^−2^ at 1.65 V although using Pt as cathode; nonetheless, the performance of the system attains about half the performance from that obtained with the SEBS-Py2O6 membrane [33]. Still, the reported performances of AEM water electrolyzers are generally high, particularly considering the circulating alkaline electrolyte. For instances, the Sustainion membranes composed of polystyrene backbone and doped with quaternized imidazolium groups allow us to reach 1 A cm^−2^ at 1.9 V using 1 M KOH while demonstrating a great stability [9]. However, Sustainion membranes work poorly in pure water, where a significant decrease in conductivity is visible, resulting in over one order of magnitude lower performance in contrast to more mature PEM water electrolyzers, i.e., reaching less than 500 mA cm^−2^ at 2 V [48,49].

Although the use of PGM-free materials in UPW operation is the most wanted option to improve AEM electrolysis competitively, new challenges arise, primarily concerning the lower conductivity of membranes and considerably slower kinetics of the catalysts at neutral pH.

In this regard, the AEMWE was tested using UPW as liquid electrolyte to unveil the effect of the spacer addition on the retention capacity of the hydroxyl ion of the membrane and ionomer. After retrieving several polarization curves, as shown in Figure 6, the cell did not display any signs of performance decay since no hysteresis could be seen; remarkably, a stable current density of 171 mA cm^−2^ could be achieved at 2 V and 60 °C with UPW feed. This result is promising since the AEMWE cell outperforms previously reported AEMWE with piperidinium-functionalized SEBS [47]. It is possible that the overall hydroxyl ion conductivity and retention could be increased by the addition of the flexible ethylene oxide spacers in the formulation developed herein [50].

The Nyquist plots of the EIS shown in Figure 6b demonstrate that although the ohmic resistance of the membrane is nearly 3.6 fold-higher in ultra-pure water operation compared to operation in the OH^−^ saturated environment, the overall charge transfer increment associated with both the anode and cathode in neutral pH is rather low (*ca.* 1.3 fold higher); thereby, it is suggested that a detrimental degradation of the catalyst does not occur in UPW, since most likely the local pH does not trigger the leaching of the earth transition elements composing the catalyst, which are commonly unstable in acid/neutral conditions.

Therefore, it could be confirmed that the SEBS-P206-based AEM/AEI is able to retain its OH^−^ conductivity at not only high temperatures, i.e., at 60 °C, in 0.1 M KOH but also in pH-neutral media, which poses a versatile option in AEM electrolysis since the possibility of eliminating KOH-based electrolytes is highly anticipated to lower operation expenditure of the technology.

However, further improvements at the cell level are necessary, namely by optimizing the PTL used at the anode side by increasing the operation temperature and understanding its effect on chemical/mechanical stability of the SEBS-P206 membrane and ionomer and also by attempting in situ strategies for increasing the long-term OH^−^ retention in UPW.

## 4. Conclusions

Novel piperidinium-functionalized flexible ethylene oxide spacer side-chain SEBS AEM was designed, synthesized, and applied as a membrane for AEMWE. In order to improve the conductivity of the membrane, a comb-shaped multi-functional flexible ethylene oxide spacer structure on the side chain was synthesized and investigated, revealing that flexible ethylene oxide side chains as spacers in the comb-like structures can promote a microphase separation in the membrane morphology and thus optimize the water uptake and enhanced ion conductivity. Moreover, having a pre-functionalized end of each side chain could also promote the IEC of the AEM and positively affect the ion conductivity so AEMs with piperidinium-end flexible ethylene oxide spacer side chains had relatively higher conductivity (25 mS cm^−1^, OH^−^ form at room temperature) and chemical stability (nearly 95% conductivity retention after in 1 M KOH for 720 h). In the next approach, pyrrolidinium was introduced as a functional group for enhancing the alkaline stability. Furthermore, SEBS-Py2O6 reached conductivity of 27.8 mS cm^−1^ (OH^−^) at RT, which is higher than SEBS piperidinium/pyrrolidinium membrane published in the past [47]. Finally, in single-cell AEMs, water electrolysis tests with PGM-free catalyst (OXYGN-N and H2GEN-M) at 60 °C achieved current densities of 171 mA cm^−2^ at 2 V cell potential and 520 mA cm^−2^ at 2 V in UPW and 0.1 M KOH, respectively. We have shown that the SEBS-P2O6 membrane for AEMWE in pure water outperforms membranes based on SEBS reported in the literature [20,33]. Additionally, the combination with of PGM-free electrodes in a AEMWE cells showed that SEBS-Py2O6 is a promising candidate for AEM water electrolysis cells.

## Data Availability

Data available on request due to privacy restrictions. The data presented in this study are available on request from the corresponding author.

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
