# Peer review of "Novel Pyrrolidinium-Functionalized Styrene-b-ethylene-b-butylene-b-styrene Copolymer Based Anion Exchange Membrane with Flexible Spacers for Water Electrolysis"

_membranes, 2023, doi:10.3390/membranes13030328_

Round 1
Reviewer 1 Report
The Authors in the manuscript have improved the stability and conductivity of the AEM by intrducing the comb like flexible ethylene oxide spacer side chain into SEBS backbone. This is good approcah to innovate in the AEM technology for neutral water electrolyzers which is economically more viable. The rationale, methodology for membrane preparation is well documented and the extensive characterization ware done to present the data for membrane characteristics. Few questions that emerge are:
1. the authors ascribe the "no change in conductivity" data in Fig. 5 with the chemical stability by moving from piperidinium to pyrrolidinium. The structural/chemical instability in KOH can come from other sources as well. Any comments?
2. It would be very helpful if the authors compared the Polarization curves of this system with already reported works.
Author Response
Thank you for your rsuggestions. There are replies to the questions.
1.Yes, the chemical stability drop may also from the radical attack. I add the possibility in the article.
2.Because my cell was tested in 0.1m KOH and with PGM-free catalyst, there are not so many papers which can have similar systems which could compare. Therefore, Before in this paper I only compared my cell results with the cell which their membranes were also SEBS based and piperidinium functionalized. However, now I have added a paragraph to the manuscript (From line 390) comparing our findings to the top outcomes in the literature.

Reviewer 2 Report
The aim of presented manuscript is to create a new type of anion exchange membrane for AWE water electrolysis and AEM-based energy systems. The new membranes have a good chemical stability and conductivity. It will be of interest for the experts in the field of membranes and membrane technologies. However, there are some comments on the manuscript.
General concept comments
1) Was the membrane removed from the HCl solution before titrating during the determination of the ion exchange capacity? Was the entire volume of the contact solution titrated, or a sample of it? It should be specified.
2) The hydrochloric acid volume of 3 mL is questionable. What was the mass of the membrane? Could the high error in the determination of the ion-exchange capacity (13-18% for samples 3-6 in the Table 1) be a result of a small mass of the membrane?
3) A set-up scheme has to be included to the Section 2.5
4) There is no need to refer to the article [35] in the Line 163, because there is no extra information about the method of the conductivity determination in it. A more thorough description of the method of determining the conductivity has to be added. Was a contact or a difference method used? Which electrodes were used?
5) How is a lower conductivity of the SEBS-Py2C6 membrane compared to the SEBS-P2O6 and SEBS-Py2O6 ones explained, given that all these membranes have comparable ion-exchange capacity values (Table 1)?
6) The term «a steady state» (Line 386-387) is not quite clear. Figure 6 shows the current-voltage curves, which is a dynamic characteristic. How, on the basis of this data, the authors assume the onset of a stationary state? It should be specified.
7) What was the purpose of using ultra-pure water as an «electrolyte at neutral ph» (Line 243)? Why not use, for example, a sodium sulfate or nitrate as an electrolyte at neutral pH? What is the resistance and the composition of this water? According to the IV-curve (Figure 6, a), this is a dilute solution of some electrolyte or electrolytes. Which electrode reactions proceed in the solution? It is obvious, that an increase in the system resistance is connected to a low concentration of electrolytes in the ultra-pure water. In addition, the presence of any anions but OH- in the solution would lead to ion-exchange reactions between the solution and the anion-exchange membrane. Do the authors take it into consideration? This part of the manuscript (3.5 AEM water electrolysis measurements) has to be detailed.
Specific comments
8) Scheme 1 has a low resolution of the Figure 1. The conditions of the synthesis and catalysts are not readable. The figure must be replaced.
9) In the Line 173 the term «swelling degree» is used while in the Table 1 – the term «swelling ratio».
10) The membranes thicknesses have to be added to Table 1.

Author Response
Reviewer 2
The aim of presented manuscript is to create a new type of anion exchange membrane for AWE water electrolysis and AEM-based energy systems. The new membranes have a good chemical stability and conductivity. It will be of interest for the experts in the field of membranes and membrane technologies. However, there are some comments on the manuscript.
General concept comments
(1) Was the membrane removed from the HCl solution before titrating during the determination of the ion exchange capacity? Was the entire volume of the contact solution titrated, or a sample of it? It should be specified.
A: Yes, the membrane was removed from HCl before titrating and it is entire volume and I added this in the manuscript.
(2) The hydrochloric acid volume of 3 mL is questionable. What was the mass of the membrane? Could the high error in the determination of the ion-exchange capacity (13-18% for samples 3-6 in the Table 1) be a result of a small mass of the membrane?
A: Yes, thank you for the advice. The membrane is around 20 mg and just to put excess amount of the HCl. After your suggestion, it seems that 3 mL is too much and next time I will try to decrease the concentration of HCl and NaOH or decrease the amount of solvent.
(3) A set-up scheme has to be included to the Section 2.5
A: I added the set-up scheme in the section.
(4) There is no need to refer to the article [35] in the Line 163, because there is no extra information about the method of the conductivity determination in it. A more thorough description of the method of determining the conductivity has to be added. Was a contact or a difference method used? Which electrodes were used?
A: I deleted the reference. It is the contact methods, and gold electrode was used.
(5) How is a lower conductivity of the SEBS-Py2C6 membrane compared to the SEBS-P2O6 and SEBS-Py2O6 ones explained, given that all these membranes have comparable ion-exchange capacity values (Table 1)?
A: We explain this in AFM part. The broader and better interconnected hydrophilic domains of SEBS-Py206 make the ion-channels much better intercalated with relatively constant size, which enables efficient ion transport
(6) The term «a steady state» (Line 386-387) is not quite clear. Figure 6 shows the current-voltage curves, which is a dynamic characteristic. How, on the basis of this data, the authors assume the onset of a stationary state? It should be specified.
A: Thank you for the comment. We have updated the manuscript with your suggestion.
We have performed a potentiostatic conditioning step prior to running the polarization curves, until reaching the steady state, i.e constant current output at every potential step.
Figure S1. (a) potentiostatic conditioning in 0.1 M KOH electrolyte; (b) potentiostatic conditioning in UPW.
(7) What was the purpose of using ultra-pure water as an «electrolyte at neutral ph» (Line 243)? Why not use, for example, a sodium sulfate or nitrate as an electrolyte at neutral pH? What is the resistance and the composition of this water? According to the IV-curve (Figure 6, a), this is a dilute solution of some electrolyte or electrolytes. Which electrode reactions proceed in the solution? It is obvious, that an increase in the system resistance is connected to a low concentration of electrolytes in the ultra-pure water. In addition, the presence of any anions but Ohi n the solution would lead to ion-exchange reactions between the solution and the anion-exchange membrane. Do the authors take it into consideration? This part of the manuscript (3.5 AEM water electrolysis measurements) has to be detailed. Specific comments
A: Thank you for the comment.
We have used neutral pH with ultra pure water as it is reported as the most anticipated solution for the AEM water electrolysers to become as competitive as PEM water electrolysers, due to the dismissing of a supporting electrolyte and the possibility of using cheap abundant materials, namely membranes and PGM-free catalysts.
For running the experiments in pH-neutral media, we have used Millipore quality water treated from a reverse osmosis system, with a resistivity of 18.2 MΩ cm at 25 °C and a TOC value below 5 ppb. Therefore, this water, UPW, owns the same quality of that used for running PEMWEs. Also, we reinforce that this experiment considers a new MEA assembly, with solely a previous activation of the membrane to include OH- ion in the polymer matrix. Then, the MEA was purged for half an hour in a stirred ultra-pure water vessel until achieving a neutral pH at the exhaust of the cell; then, the MEA was placed in the test bench for running the break in and polarization curve in pH=7. We therefore expect that there is no additional side reaction due to dissolved ionic species other than OH-, i.e we expect that solely OER and HER had occurred. We opted not to use other ion dissolved electrolytes since most reported AEM electrolyzers operate in carbonate- or hydroxide-based electrolyte that end up masking the stability challenges associated to the ionomer and membranes in the alkaline oxidative environment, mainly due to unwanted nucleophilic substitution and Hoffmann elimination of the positive functional groups at the polymer matrix (https://doi.org/10.1002/anie.20130809). Additionally, we looked forward to analyse the effect of the added spacers on the retention of OH- in diluted solutions and we believe it could easily be assessed by running the AEMWE ultra-pure water conditions. Similarly, the interaction between the solid electrolyte ionomer/membrane and PGM-free catalysts was of special interest for us to be analysed in UPW, as commonly the catalyst degradation is rapidly triggered. Nonetheless, the addition of spacers appears to have had a positive effect as we have noticed that after a 20 h chronopotentiometry at 200 mA·cm-2, the performance was actually increasing and the polarization curve at end of life shows low ohmic drop and increased mass transport (Figure 2, below), which unveils the membrane herein developed owns promising features for further application in AEMWEs in low alkalinity (we will show this in our future work).
We have noticed, that upon running several polarization curves (i.e, 5 I/V curves) there was no sign of performance decay and an increase of conductivity could be perceived in comparison to other works reporting AEMWE operating with UPW.
Fig.2 (a) chronopotentiometry at 200 mA cm-2, 60 °C for AEMWE running in UPW using SEBS-PY2O6 and PGM-free catalysts; (b) beginning of life and end of life I/V curves for AEMWE in UPW.
(8) Scheme 1 has a low resolution of the Figure 1. The conditions of the synthesis and catalysts are not readable. The figure must be replaced.
A: I have exchanged it.
(9) In the Line 173 the term «swelling degree» is used while in the Table 1 – the term «swelling ratio».
A: Thank you. I have corrected the mistake.
(10) The membranes thicknesses have to be added to Table 1.
A: I have added in Table 1.

Reviewer 3 Report
The overall writing is indeed interesting. The reviewer only curious how you determine diffusion coefficient in the spacer fabric? As it is known that the diffusion coefficient in the spacer fabric is different from those passing through the membrane surface. Hope the reviewer can clarify.
Author Response
Thank you for your comments and suggestions. There is the answer to the suggestion.
1.In membranes, we always use conductivity to describe the diffusion and movements of the ions. In this paper, we measure conductivity through plane and it is only the hydroxide diffusion through the membrane. Because now characterization methods are still limited, we cannot measure the conductivity of the passing through the surface and in spacer fabric separately.
Round 2
Reviewer 2 Report
I thank the authors for their detailed responses.
Author Response
Thank you for reviewing my manuscript